# Does family life education influence attitudes towards sexual and reproductive health matters among unmarried young women in India?

**Niharika Tripathi**⊙*

ICMR-National Institute of Medical Statistics (NIMS), Ansari Nagar, New Delhi

* niharika.t2010@gmail.com

## Abstract

### Introduction

Inadequate efforts towards meeting the sexual and reproductive health needs of adolescents and young people, who disproportionately share the burden of unwanted pregnancies, poor maternal and child health outcomes, risks of RTI/STI and HIV/AIDS, increase the risk of losing much of the progress made towards the Millennium Development Goals over the last decade, particularly in the context of low-and-middle-income countries like India.

### Data and methods

Using the nationally representative data on 160551 unmarried young women aged 15–24 years from the District Level Reproductive and Child Health Survey (DLHS: 2007–2008) in India, this research evaluated the demographic and socioeconomic differentials in the access to family life/sex education (FLE) among youth in India. Using the adjusted multiple logistic regression models, the association between access to family life/sex education and attitudes towards a range of sexual and reproductive health matters among young unmarried Indian women were investigated.

### Results

Less than half of the unmarried young women had received some form of FLE (48 percent) in India. However, there were substantial demographic and socioeconomic variations in their access to FLE, as relatively less educated women from the poorest wealth quintiles, religious and social minorities (Muslims, Scheduled Castes/Scheduled Tribes) were significantly less likely to receive FLE as compared to other women. Importantly, the likelihood of holding favourable/positive attitudes towards reproductive processes, knowledge and discussion of contraceptive methods, precise awareness about the transmission pathways of RTIs/STIs and HIV/AIDS was significantly higher among those women in India who had access to FLE.

**Data Availability Statement:** The dataset is available in the public domain for research use and it can be downloaded from the website of the

International Institute of Population Sciences (IIPS) at: http://rchiips.org/PRCH-3.html.

**Funding:** The authors have no support or funding to report.

**Competing interests:** The authors have declared that no competing interests exist.

## Conclusion

The present research underscores the protective role of family life education towards improving the sexual and reproductive life experiences of young people. It further underscores the vital need to implement a comprehensive and culturally appropriate programme of family life education in order to meet the sexual and reproductive health demands of the adolescents and young people in India.

## 1. Introduction

Currently, the world is witnessing one of the most profound demographic changes in the history of humankind. About 41 percent of the world's population is younger than 25 years, with a majority of them (89 percent) residing in the less developed countries [1]. Furthermore, around 24 percent of the world's population is between 10–24 years of age. Approximately 1.3 billion people are adolescents (10–19 years), with almost 89 percent of them living in the developing countries [1,2]. This young brigade, often referred to in demographic literature as the 'window of opportunity', potentially represents the future generation of parents, teachers, technocrats, managers and upcoming leaders in the globalising world. However, these cohorts of adolescents and young people seem to be living in a complex web of an increasingly urban life style associated with dwindling traditional family ties, escalating unemployment, juvenile violence, youth suicide, early menarche/puberty, early sexual debut, non-consensual sexual encounters, rising age at marriage, unintended parenthood, clandestine abortions and high risk of RTIs/STIs and HIV/AIDS [3–9]. These unprecedented transitions in the socioeconomic, demographic and health spheres of modern life call for specific concerted efforts towards comprehensive investments in the fields of education, health and economic opportunities, both across and within countries. Such inputs will strengthen the earlier efforts (prenatal care, delivery and postnatal care, immunisation care, nutritional interventions, primary and secondary education programmes etc.) undertaken across several developing countries during the first decade of life of these young adolescents, who grew up against the backdrop of International Conference of Population and Development 1994 and the Millennium Development Goals 2000 [10]. These positive initiatives may have reinforced the young adolescent's need to escape the intergenerational transmission of the vicious circle of poverty, hunger, illiteracy, ill-health and unemployment, particularly in the developing world [11].

However, the paucity of relevant data and empirical evidence, particularly in the developing world, largely restricted the extent of progress which could have otherwise been made to address the range of adverse economic, social, health and educational concerns routinely confronted by adolescents and young people across the globe [3,12]. It is noteworthy that young adolescents contributed to 16 million births worldwide, accounting for around 11 percent of all births in 2008. Interestingly, 95 percent of these births occurred in the developing countries [13]. The highest incidence of adolescent fertility rates has been reported in Sub-Saharan Africa and in the South Asian countries. Research based on the data of 56 countries indicates that young women aged 15–19 years belonging to the economically weaker groups, were at three times the elevated risk of pregnancy during adolescence than their richer counterparts. These young and economically poor adolescent women were at about five-time higher risk of death from pregnancy related causes and their offspring also experienced substantially higher risk of death. The episodes of adolescent pregnancy also lead to other multiple negative outcomes including school dropouts and lower educational attainment restricting women's

lifetime earnings [14–16]. Though pregnancy and births among adolescent between 10–14 years are limited, still they vary from 0.3 to 12 percent in parts of Sub-Saharan Africa. Such concerning patterns among adolescents are fuelled by declining pubertal age, low age at marriage, large spousal age gap and gendered norms discouraging the negotiating power to use contraception particularly among stable marital unions. Furthermore, gendered societal norms that disempower women perpetuate the risks of pregnancy and sexually transmitted infections (STIs) including HIV/AIDS among young adolescent women. In addition, about two to four million adolescents in developing countries experience clandestine and unsafe abortions annually. About 11 percent of young women 15–19 years reported the unmet need for contraception, while seven in 10 individuals aged 15–24 years contracted STIs [13].

India is one of the youngest nations accounting for the largest national population of adolescents (243 million) in the world followed by China (207 million), United States (44 million), Indonesia and Pakistan (41 million each) [17]. About 327 million Indians (30 percent) are between 10–24 years of age and approximately 70 percent are below 35 years [18]. Despite being relatively more educated, urbanised and healthier than earlier generations, these cohorts of young India of the 21st century face substantially complex economic, social and health hazards of the kind probably never witnessed before. The inadequate knowledge and agency on the part of Indian youth to make informed sexual and reproductive health (SRH) choices also threaten serious risks related to their sexual and reproductive health. The National Population Policy (2000), the National AIDS Prevention and Control Policy (2002), the National Youth Policy (2003), the Reproductive and Child Health Programmes (1 and 2) during 1997 and 2005 reiterate the commitment of the Indian government to recognise the urgent need to address the sexual and reproductive health issues of adolescents and young people. Moreover, the Government of India is also a signatory to the ICPD and ICPD+5 Programme of Action that underscores its obligation to protect and promote the rights of adolescents to the enjoyment of an attainable standard of health. Furthermore, implementation of the Adolescent Reproductive and Sexual Health Strategy (ARSH) (2005–2013), and the national adolescent health programme "Rastriya Kishor Swasthya Karyakram—RKSK" (2014- present) reaffirmed the union government's resolution and political will to empower and strengthen adolescent health and well-being. These programmes have helped to bridge important gaps in access to quality health care services among the adolescent and youth population, through introduction of Adolescent Friendly Health Clinics (AFHCs), community health workers (ANM and ASHA workers), male and female counsellors, and peer educators. Other important initiatives under RKSK included celebration of Adolescent Health Day (AHD), peer education for greater community level engagement among adolescents, weekly iron folic acid supplementation programme to address higher prevalence of anaemia, and menstrual hygiene skills for promoting healthy and safe sanitary practices in an environmentally friendly manner [19,20].

However, previous research on an array of sexual and reproductive health (SRH) issues typically related with adolescents and youth in India reveal some worrying realities [12,18,21–23]. Generally, young men as compared to women, tend to adhere to casual sexual behaviour marked by low condom use (less than 30 percent) during sex with casual partners. This leads to large numbers of youth 15–24 years suffering from sexually transmitted infections (STIs) including HIV/AIDS and accounts for about 35 percent of the total AIDS burden in India [24]. Furthermore, awareness levels about STIs remained limited among Indian youth, as only 29 percent had heard about it. Intriguingly, less than 20 percent of Indian youth were aware that STI increases the risk of HIV/AIDS. The substantial gender inequality in Indian societies economically and socially disempowers young women, increasing their risk of falling prey to forced sexual activities and violence, and elevates the risk of HIV/AIDS. Consequently, women share around 38 percent of the all India HIV/AIDS burden, with the prevalence of HIV/AIDS

among young women far exceeding that same among young men. The rising trend of injecting drug use (IDUs) among young people in India, particularly along the North-Eastern part of the country has added to the incidence of HIV/AIDS. Compounding this problem is the fact that the utilisation of sexual and reproductive health services by Indian youth is very limited.

Available evidence explicitly brings out the fact that the adolescent and youth populations often adopt risky behaviours largely due to inadequate and inaccurate information about a range of sexual and reproductive health issues including physical attributes, sexuality, and the ramifications of unprotected sexual encounters in the Indian context. In addition, they often lack essential skills to resist unsafe sexual encounters. Furthermore, injecting drug use along with tender age and gender inequalities elevates the risk of contracting RTIs/STIs and HIV/AIDs. This worrying state of affairs among adolescents and young people can be averted by imparting to them correct knowledge and enhancing their awareness of sexual and reproductive health matters. In order to improve the SRH among adolescents and young people, one of the critical recommendations from the ICPD+25 (2019) resolution was to ensure 'access for all adolescents and youth, especially girls, to comprehensive and age-responsive information, education and adolescent-friendly comprehensive, quality and timely services to be able to make free and informed decisions and choices about their sexuality and reproductive lives, to adequately protect themselves from unintended pregnancies, all forms of sexual and gender-based violence and harmful practices, sexually transmitted infections, including HIV/AIDS, to facilitate a safe transition into adulthood' [25].

Experiences of family life education (FLE) -that mainly seek to impart necessary information and skills to understand human sexuality and learn healthy behaviours to celebrate the transition from childhood to adulthood- have received mixed responses across countries. However, some of the studies have documented positive SRH behavioural changes through family life education among adolescents and youth, like improved understanding of SRH issues, increased use of contraception, delayed sexual debut, partner selection, avoiding unsafe abortion, better menstrual hygiene and lowering risk of STIs and HIV/AIDS [26–33]. Nevertheless, there is no clear understanding regarding the association between access to family life education and reproductive health behaviours among youth in India [34]. In fact, studies on the scope of implementing family life education per se in India and in its states are extremely limited [35]. Throughout our literature search, we did not come across any nationally representative study that examined the influence of family life education on SRH behaviour of the adolescent and youth population in India.

Therefore, the present study serves two crucial objectives. First, we assess the extent of access to family life education (FLE) as per the socioeconomic and demographic characteristics of young unmarried women aged 15–24 years in India. Second, we investigate the association between access to FLE and experience/awareness/attitude towards sexual and reproductive health matters among young unmarried women in India.

## 2. Data and methods

The data for the present study comes from the third round of the nationally representative District Level Household and Facility Survey (DLHS-3) of India conducted from December 2007 to December 2008 [21]. The DLHS-3 is one of the largest ever demographic and health surveys carried out in India with a sample size of 7,20,320 households and 6,43,944 ever married women across 34 States and Union Territories covering 601 districts of the country. It provides national/state/district level estimates to monitor the maternal and child health, family planning and reproductive health indicators. For the first time, DLHS-3 interviewed unmarried women aged 15-24years (1,60,551) to provide information on family life/sex education, awareness

about reproductive health, contraception and HIV/AIDS etc. DLHS-3 adopted a multi-stage stratified systematic sampling design that resulted in national/state/district representative samples after applying sampling weights to adjust for complex survey design.

## 2.1 Outcome variables

The study employed five continuous outcome variables to understand the awareness, attitudes and experiences related to the sexual and reproductive health practices among unmarried young women aged 15–24 years in India (**Table 1**). First, an additive index of menstruation was generated using four different items (0–4) related to experiences and management of menstruation, with a relatively higher score suggesting better menstrual health and hygiene management. Second, an additive index of reproduction was generated using five different items (0–5) related to awareness about reproductive health matters, with a relatively higher score suggesting enhanced understanding and awareness related to reproductive health matters. Third, an additive index of contraception was generated using three different items (0–3) related to awareness about family planning methods, with a relatively higher score indicating improved awareness and knowledge about family planning methods. Fourth, an additive index of reproductive tract infection/ sexually transmitted diseases (RTI/STI) was generated using seven different items (0–7) related to awareness about transmission of RTI/STI related matters, with a relatively higher score indicating better understanding and know-how about RTI/STI issues. Fifth, an additive index of HIV/AIDS was generated using seven different items (0–7) related to awareness about transmission of RTI/STI related matters, with a relatively higher score indicative of enhanced awareness and proficiency about family planning methods [36]. In addition, we have also used family life education (binary variable: yes = 1; no = 0) as a dependent variable in one set of cross-tabulations and a binary logistic regression model. In the bivariate analyses, we separately used individual variables/items related to different measures of knowledge and awareness of reproductive health matters to highlight the variability across the access to family life education among young unmarried women.

## 2.2 Exposure variables

The main exposure variable in the study was access to family life education (FLE) among unmarried young women aged 15–24 years in India. The DLHS-3 administered the following question to elicit basic understanding related to FLE among unmarried young women 15–24 years–*"Are you aware of the family life education/sex education programme*, i.e., *about bodies*, *growing up*, *male-female relationship*, *and sexual matters*?"*. In order to evaluate the access to FLE, the DLHS-3 asked unmarried young women aged 15–24 years- "*Have you ever received any family life/sex education*?", and further these respondents were asked- "*If yes*, *where did you receive that*?" The second question mentioned above was used to generate the binary variable of FLE. We also used other pertinent demographic and socioeconomic variables that might have confounded the association between access to FLE and attitudes toward SRH among respondents. These included current age of women, years of women schooling, women employment status, place of residence, caste/social groups, religious groups, wealth status and province/state.

## 2.3 Data analysis

The study used cross tabulations to examine the bivariate distribution of the response variables across a set of explanatory variables. Chi-Square tests were performed to check the statistically significant differences in outcome variables by selected demographic and socioeconomic attributes. We used the multiple logistic regression model to examine the determinants of FLE among young unmarried women in India. In addition, multiple linear regression models were

**Table 1. Description of the study variables.**

| Sl. No. | Characteristics | Description of study variables |
|---|---|---|
| | Dependent variables | |
| 1 | Index of Menstruation (0–4) with mean = 1.33 ± SD 0.57 | 1.1 Experience of any menstruation related problem in last three months? (Yes = 1, No = 0) |
| | | 1.2 Methods of protection during menstrual period-cloth (Yes = 1, No = 0) |
| | | 1.3 Methods of protection during menstrual period-locally prepared napkins (Yes = 1, No = 0) |
| | | 1.4 Methods of protection during menstrual period-sanitary napkins (Yes = 1, No = 0) |
| 2 | Index of Reproduction (0–5) with mean = 2.00 ± SD 1.21 | 2.1 Is sex-determination before birth possible? (Yes = 1, No = 0) |
| | | 2.2 Pregnancy possible after kissing/hugging? (Yes = 0, No = 1) |
| | | 2.3 Women must bleed after debut of sexual intercourse (Yes = 0, No = 1) |
| | | 2.4 Women can get pregnant on first intercourse (Yes = 1, No = 0) |
| | | 2.5 High risk of pregnancy for women by sexual intercourse during halfway between her periods (Yes = 1, No = 0) |
| 3 | Index of Contraception (0–3) with mean = 2.09 ± SD 0.69 | 3.1 Know any contraceptive method (Yes = 1, No = 0) |
| | | 3.2 Know any modern contraceptive method (Yes = 1, No = 0) |
| | | 3.3 Had discussion on contraception (Yes = 1, No = 0) |
| 4 | Index of Reproductive tract infection/sexually transmitted infection (RTI/STI) (0–7) with mean = 0.69 ± SD 1.25 | 4.1 Heard about RTI/STI (Yes = 1, No = 0) |
| | | 4.2 Source of RTI/STI Transmission: Unsafe delivery (Yes = 1, No = 0) |
| | | 4.3 Source of RTI/STI Transmission: Unsafe abortion (Yes = 1, No = 0) |
| | | 4.4 Source of RTI/STI Transmission: Unsafe IUD insertion (Yes = 1, No = 0) |
| | | 4.5 Source of RTI/STI Transmission: Unsafe sex with persons with many sex partner (Yes = 1, No = 0) |
| | | 4.6 Source of RTI/STI Transmission: Unsafe sex with sex worker (Yes = 1, No = 0) |
| | | 4.7 Source of RTI/STI Transmission: Unsafe sex with homosexuals (Yes = 1, No = 0) |
| 5 | Index of HIV/AIDS (0–7) with mean = 2.36 ± SD 1.86 | 5.1 Heard about HIV/AIDS (Yes = 1, No = 0); |
| | | 5.2 Source of HIV/AIDS Transmission: Unsafe sex with homosexual (Yes = 1, No = 0); |
| | | 5.3 Source of HIV/AIDS Transmission: Unsafe sex with persons with many sex partner (Yes = 1, No = 0); |
| | | 5.4 Source of HIV/AIDS Transmission: Unsafe sex with sex worker (Yes = 1, No = 0); |
| | | 5.5 Source of HIV/AIDS Transmission: Unprotected sex with HIV/AIDS person (Yes = 1, No = 0); |
| | | 5.6 Source of HIV/AIDS Transmission: Infected mother to child (Yes = 1, No = 0); |
| | | 5.7 Source of HIV/AIDS Transmission: Transfusion of infected blood (Yes = 1, No = 0); |

(*Continued*)

**Table 1.** (Continued)

| Sl. No. | Characteristics | Description of study variables |
|---|---|---|
| 6 | Independent variables | Family life education (Yes = 1, No = 0) |
| 7 | | Age group (in years) (15-19y = 1, 20-24y = 2) |
| 8 | | Residence (Rural = 1, Urban = 2) |
| 9 | | Years of schooling (Non-literate = 1, 1–5 years = 2, 6–9 years = 3, 10 years and above = 4) |
| 10 | | Employment status (Not working = 1, Agriculture = 2, Manual = 3, Non-manual = 4) |
| 11 | | Religion (Hindu = 1, Muslim = 2, Christian = 3, Others-4) |
| 12 | | Caste/tribe (Scheduled caste = 1, Scheduled tribe = 2, Other backward class = 3, Others = 4) |
| 13 | | Wealth index (Lowest = 1, Second = 2, Middle = 3, Fourth = 4, Highest = 5) |
| 14 | | States- All Indian states and union territories (35) |

Five additive index scale were generated using selected items, as mentioned above, to measures the sexual and reproductive health related knowledge/awareness/experiences among young unmarried women (15-24y) in India.

fitted to investigate the adjusted effects of access to FLE on experience/awareness/attitude on a range of sexual and reproductive health matters among unmarried young women, after controlling for demographic and socioeconomic confounding variables. The Wald test was used to check for the power of statistical significance of the exposure variables in the multiple logistic regression models. Throughout the paper, family life/sex education terms have been used interchangeably. All the analyses were performed using statistical weights to adjust for the complex survey design in STATA 13.1 version.

## 2.4 Ethics statement

The study was based on a secondary dataset with no identifiable information on the survey participants. The dataset is available in the public domain for research use and, hence, no approval was required from any institutional review board. The data can be downloaded from the website of the International Institute of Population Sciences (IIPS) at: http://rchiips.org/PRCH-3.html & https://www.iipsindia.ac.in/node/194. The data for the current study was downloaded from the aforementioned website after receiving permission.

## 3. Results

### 3.1 Descriptive results

**3.1.1 Family life education by demographic and socioeconomic attributes.** Table 2 presents the percent distribution of the sampled young unmarried women by selected demographic and socioeconomic characteristics. In addition, it also demonstrates the differentials in the access to family life education (FLE) by selected demographic and socio-economic characteristics among young unmarried women in India. Data suggest that close to three-fourth of the sampled young unmarried women were 15–19 years old, predominantly residing in the rural areas, about half had received secondary level education, three-fourths were not engaged in any gainful economic-activity, two-third were Hindus, one-third belonged to the Scheduled Caste and Scheduled Tribe communities, and every four out of ten young women belonged to

**Table 2. Percent distribution of unmarried young women (15–24 years) who received family life education (FLE) by selected demographic and socioeconomic characteristics in India, DLHS-3, 2007–2008.**

| Characteristics | Received Family life education (FLE) (%) | Percent distribution of unmarried women (%) | Sample size (N) | P-values |
|---|---|---|---|---|
| **Total sample** | **48.5** | **100.0** | **1,60,551** | |
| **Age group (in completed years)** | | | | |
| 15-19y | 46.0 | 73.5 | 1,20,586 | |
| 20-24y | 55.8 | 26.5 | 39,965 | 0.000 |
| **Residence** | | | | |
| Rural | 43.4 | 54.8 | 1,17,428 | |
| Urban | 54.8 | 45.2 | 43,123 | 0.000 |
| **Years of schooling** | | | | |
| No education | 16.0 | 8.5 | 15,269 | |
| 1–5 years | 21.2 | 10.7 | 19,226 | |
| 6–9 years | 42.9 | 35.3 | 59,900 | |
| 10 years or above | 65.5 | 45.5 | 66,156 | 0.000 |
| **Employment status** | | | | |
| Not working | 51.3 | 73.8 | 1,13,977 | |
| Agriculture | 34.6 | 12.3 | 25,462 | |
| Manual | 37.9 | 8.5 | 13,612 | |
| Non-manual | 59.9 | 5.3 | 7,500 | 0.000 |
| **Religion** | | | | |
| Hindu | 48.5 | 69.7 | 1,13,754 | |
| Muslim | 42.6 | 15.3 | 20,812 | |
| Christian | 57.5 | 7.5 | 12,817 | |
| Other religions | 52.0 | 7.6 | 13,168 | 0.000 |
| **Caste group** | | | | |
| Scheduled Caste | 44.9 | 16.0 | 26,577 | |
| Scheduled Tribe | 44.8 | 17.7 | 32,669 | |
| Other Backward Class | 46.7 | 36.7 | 56,703 | |
| Other Caste | 55.0 | 29.6 | 44,602 | 0.000 |
| **Wealth index** | | | | |
| Lowest | 25.0 | 8.6 | 17,955 | |
| Second | 32.9 | 11.9 | 23,948 | |
| Middle | 40.6 | 17.6 | 32,823 | |
| Fourth | 50.2 | 25.2 | 40,748 | |
| Highest | 61.9 | 36.7 | 45,077 | 0.000 |

Chi-square test is used to examine statistically significant difference in the family life education across selected demographic and socioeconomic characteristics among unmarried young women 15-24y.

an economically weaker background. Overall, close to 50 percent of the unmarried young women aged 15–24 years had received FLE, with marked variations across demographic and socioeconomic characteristics in India. Access to FLE was seen to be relatively higher among young unmarried women who were 20–24 years old, residing in urban areas, having secondary level or above educational attainment, engaged in non-manual occupations, were Christians, from other caste groups, and belonged to economically better-off households.

**3.1.2 Access to family life education and experiences/attitudes about sexual and reproductive health behaviour.** DLHS-3 elicited important information related to the experience of menstruation related problems among young unmarried women, during the last three

**Table 3. Percent distribution of unmarried women (15–24 years) who reported the menstruation experiences by their FLE status in India, DLHS-3, 2007–2008.**

| Characteristics | Received FLE | |
|---|---|---|
| | Yes | No |
| Experienced any menstruation related problems in last 3 months*** | 23.4 | 20.9 |
| Methods of protection during menstrual period- cloth*** | 67.4 | 82.7 |
| Methods of protection during menstrual period- locally prepared napkins*** | 11.5 | 7.1 |
| Methods of protection during menstrual period- sanitary napkins*** | 38.3 | 18.7 |

Results from Pearson's Chi Square test indicating association of menstrual problems with access to family life/sex education among young unmarried women

***p<0.001). ***p<0.001; **p<0.05; *p<0.10.

months from the date of survey. It also collected data related to the different methods used by young unmarried women for the management of menstruation including cloth, locally prepared napkins and sanitary napkins. Data suggest that among women who received FLE, about 23 percent reported having experienced some kind of menstruation related problem in the last three months before the survey date, as compared to 21 percent among women who had not received any FLE (**Table 3**). Furthermore, relatively large proportions of young women who did not receive FLE reported using cloth as the means of protection during their menstrual periods. In comparison, young women who received FLE were found to have used locally prepared napkins or sanitary napkins in greater proportion than their counterparts.

**Table 4** depicts the level of awareness about various reproductive health matters among young unmarried women. In general, the knowledge and awareness about reproductive health issues among young unmarried women was extremely limited with large demographic and socioeconomic differentials. A substantial gap regarding knowledge of reproductive health matters was observed between women who received FLE and their counterparts who did not receive any FLE. On an average, a relatively higher proportion of women who received FLE (68 percent) reported that '*it is possible to know the sex of the baby before the baby is born*' than their counterparts. Furthermore, nearly eight out of every ten women were reportedly aware of the unlikelihood that '*pregnancy can occur after kissing or hugging*'. Around 23 percent of women with FLE had correct knowledge about the fact that it was not always necessary that '*a woman has to bleed when she has sexual intercourse the very first time*', as compared to 18 percent women without any FLE. Correct knowledge regarding the facts that '*a woman can get pregnant the very first time she has sexual intercourse*', and '*a woman is more likely to get*

**Table 4. Percent distribution of unmarried women (15–24 years) who reported the information on relevant reproduction related matters by their FLE status in India, DLHS-3, 2007–2008.**

| Characteristics | Received FLE | |
|---|---|---|
| | Yes | No |
| Is sex-determination before birth possible *** | 67.7 | 57.5 |
| Pregnancy possible after kissing/hugging*** | 18.8 | 33.3 |
| Women must bleed after debut of sexual intercourse*** | 76.9 | 82.1 |
| Women can get pregnant on first intercourse*** | 28.8 | 21.5 |
| High risk of pregnancy for women by sexual intercourse during halfway between her periods*** | 24.8 | 16.0 |

Results from Pearson's Chi Square test indicating association of reproduction related matters with access to family life/sex education among young unmarried women

***p<0.001). ***p<0.001; **p<0.05; *p<0.10.

**Table 5. Percent distribution of unmarried women (15–24 years) according to knowledge/awareness of contraceptive methods by their FLE status in India, DLHS-3, 2007–2008.**

| Characteristics | Received FLE | |
|---|---|---|
| | **Yes** | **No** |
| Know any contraceptive method*** | 97.7 | 90.0 |
| Know any modern contraceptive method*** | 97.7 | 89.9 |
| Had discussion on contraception*** | 34.1 | 12.4 |

Results from Pearson's Chi Square test indicating association of knowledge/awareness of contraceptive methods with access to family life/sex education among young unmarried women

***p<0.001). ***p<0.001; **p<0.05; *p<0.10.

*pregnant if she has sexual intercourse half way between her periods*' was relatively higher among young women who received FLE, as compared to their counterparts.

Table 5 presents awareness about contraceptive methods among unmarried women by their FLE status. About 98 percent of women who received FLE reported their awareness of any contraceptive method, and also about any modern contraceptive method. Interestingly, it was seen it emerged that while the awareness of contraceptive methods was quite high among the respondents, discussion on these important issues was minimal, that is, young unmarried women rarely discussed issues of contraception with their parents/friends/relatives etc. However, it was found that a relatively sizeable proportion of young unmarried women with FLE (34 percent) discussed about different methods of contraception in comparison to their counterparts (12 percent) who had received no FLE. This clearly demonstrates a vital need to educate and train unmarried young women on SRH issues so that they can freely discuss, evaluate and dispel ambivalent views/myths about the available contraceptive options and the side effects of different contraceptive methods.

The knowledge and awareness regarding reproductive tract infection and sexually transmitted infection (RTI/STI) and its transmission was quite low among unmarried young women in India (Table 6). Among young unmarried women who received FLE, around 46 percent had ever heard about RTI/STI as compared to only 21 percent among respondents who did not

**Table 6. Percent distribution of unmarried women (15–24 years) according to knowledge/awareness about RTI/STI matters by their FLE status in India, DLHS-3, 2007–2008.**

| Characteristics | Received FLE | |
|---|---|---|
| | **Yes** | **No** |
| **Heard about RTI/STI*** | | |
| Yes | 46.4 | 21.0 |
| No | 53.7 | 79.0 |
| **Source of RTI/STI Transmission?** | | |
| Unsafe delivery*** | 22.5 | 20.7 |
| Unsafe abortion*** | 18.1 | 16.1 |
| Unsafe IUD insertion*** | 14.9 | 12.1 |
| Unsafe sex with persons with many sex partner*** | 70.7 | 66.1 |
| Unsafe sex with sex worker*** | 33.5 | 33.4 |
| Unsafe sex with homosexuals*** | 21.8 | 17.3 |

Results from Pearson's Chi Square test indicating association of knowledge/awareness about RTI/STI matters with access to family life/sex education among young unmarried women

***p<0.001). ***p<0.001; **p<0.05; *p<0.10.

receive any FLE. Knowledge about the sources of transmission of RTI/STI was quite low among young unmarried women. Less than one-fourth of the respondents who received FLE were aware that RTI/STI could be transmitted as a result of unsafe delivery, unsafe abortion, unsafe IUD insertion and having unsafe sex with homosexuals. Almost two-thirds of respondents with FLE were aware that the source of transmission of RTI/STI could be unsafe sex with persons who had multiple partners. Furthermore, about one-third of young unmarried women who received FLE reported that they had the knowledge that RTI/STI could be transmitted through unsafe sex with sex workers. However, this scenario was considerably worse with regard to knowledge about RTI/STI and modes of its transmission among young unmarried women without any FLE.

Table 7 illustrates the extent of awareness and knowledge regarding HIV/AIDS and sources of its transmission among unmarried young women in India by their FLE status. Majority of young unmarried women with FLE (91 percent) had ever heard about HIV/AIDS in comparison to their counterparts (68 percent). The knowledge regarding various sources of its transmission was also investigated. More than two-thirds of unmarried young women with FLE reported that HIV/AIDS could be transmitted through transfusion of infected blood. Six in every ten women with FLE reported that the source of transmission of HIV/AIDS could be having unsafe sex with persons who had multiple partners. Unprotected sex with persons infected with HIV/AIDS and transmission from infected mother to child was reported by 42 percent and 43 percent respectively among young unmarried women with FLE. Almost one-third of the respondents with FLE reported that they were aware that HIV/AIDS could be transmitted due to unsafe sex with sex workers and only 16 percent of respondents who received FLE knew that unsafe sex with homosexuals was a source of transmission of HIV/AIDS. However, it must be noted that on all dimensions of awareness/knowledge and modes of transmission of HIV/AIDS, unmarried young women without any FLE fared dismally.

## 3.2 Multivariate results

As regard access to FLE among unmarried young women in India, results from bivariate analyses indicated marked differences across a range of sexual and reproductive health indicators.

**Table 7. Percent distribution of unmarried women (15–24 years) according to knowledge/awareness about HIV/AIDS by their FLE status in India, DLHS-3, 2007–2008.**

| Characteristics | Received FLE | |
|---|---|---|
| | Yes | No |
| **Heard about HIV/AIDS**\*\*\* | | |
| Yes | 91.2 | 67.6 |
| No | 8.8 | 32.4 |
| **Source of HIV/AIDS Transmission?** | | |
| Unsafe sex with homosexual\*\*\* | 16.3 | 11.7 |
| Unsafe sex with persons with many sex partner\*\*\* | 64.6 | 57.3 |
| Unsafe sex with sex worker\*\*\* | 31.0 | 26.4 |
| Unprotected sex with HIV/AIDS person\*\*\* | 41.7 | 34.8 |
| Infected mother to child\*\*\* | 43.2 | 33.4 |
| Transfusion of infected blood\*\*\* | 70.6 | 61.7 |

Results from Pearson's Chi Square test indicating association of knowledge/awareness about HIV/AIDS with access to family life/sex education among young unmarried women

\*\*\*p<0.001). \*\*\*p<0.001; \*\*p<0.05; \*p<0.10.

We used the multiple logistic regression model to examine the demographic and socioeconomic determinants of family life education, considering the dichotomous nature of the dependent variable (received FLE = 1; not received FLE = 0). In addition, we fitted five set of multiple linear regression models considering the continuous nature of the dependent variables, to examine the association between access to FLE and experiences/attitudes on a range of sexual and reproductive health behaviour among unmarried young women aged 15–24 years after adjusting for selected socioeconomic, demographic and contextual factors.

Table 8 presents the adjusted odds ratio from the logistic regression model, predicting the likelihood of unmarried young women aged 15–24 years in India receiving any FLE. The odds of receiving FLE was relatively more among women aged 20–24 years as compared to those in

**Table 8. Adjusted odds ratio from multiple logistic regression model predicting the likelihood of receiving FLE among unmarried women (15–24 years), India, DLHS-3, 2007–2008.**

| Characteristics | Odds ratio | P-values |
|---|---|---|
| **Age Group** | | |
| 15-19y[R] | 1.00 | |
| 20-24y | 1.04 | 0.006 |
| **Residence** | | |
| Rural[R] | 1.00 | |
| Urban | 0.97 | 0.019 |
| **Years of schooling** | | |
| Non-Literate[R] | 1.00 | |
| 1–5 years | 1.28 | 0.000 |
| 6–9 years | 3.47 | 0.000 |
| 10 years or above | 7.84 | 0.000 |
| **Employment Status** | | |
| Not working[R] | 1.00 | |
| Agriculture | 0.92 | 0.000 |
| Manual | 0.99 | 0.926 |
| Non-manual | 1.15 | 0.000 |
| **Religion** | | |
| Hindu[R] | 1.00 | |
| Muslim | 0.93 | 0.000 |
| Christian | 1.37 | 0.000 |
| Others | 1.07 | 0.008 |
| **Caste/Tribe** | | |
| Scheduled Caste[R] | 1.00 | |
| Scheduled Tribe | 0.93 | 0.001 |
| Other Backward Classes | 0.99 | 0.662 |
| Others | 0.99 | 0.500 |
| **Wealth Index** | | |
| Lowest[R] | 1.00 | |
| Second | 1.19 | 0.000 |
| Middle | 1.35 | 0.000 |
| Fourth | 1.55 | 0.000 |
| Highest | 1.80 | 0.000 |

Dependent variable is *access to family life education*- 'yes' is coded as '1' ad 'no' is coded as '0'.; Regression model is mutually adjusted for state of residence.

the 15–19 years age group. Consistent with the bivariate results, as the years of schooling or the level of education increased, the likelihood of women receiving FLE also improved. Similar positive patterns were observed regarding the wealth quintile and likelihood of receiving FLE. Women belonging to the richer/richest wealth quintiles were more likely to receive FLE as compared to their poorer counterparts.

Table 9 presents the adjusted linear association of access to family life education with five selected indicators of sexual and reproductive health behaviours among young unmarried women in India including index of menstruation, index of reproduction, index of contraception, index of RTI/STI, and index of HIV/AIDS, after controlling for selected demographic and socioeconomic factors. Data suggest that access to FLE had a statistically significant and positive association with index of menstruation among young unmarried women, after controlling for other demographic and socioeconomic factors. For instance, a one unit increase in the access to FLE had a 0.06 increase in the index of menstruation among young women, confirming the positive effect of family life education on menstrual hygiene. In addition, educational status, place of residence, caste, religion, and wealth status were also significantly associated with the menstrual health of young unmarried women in India. Furthermore, it emerged that access to FLE had a statistically significant and positive association with the index of reproduction after controlling for other demographic and socioeconomic factors. For instance, a unit increase in the access to FLE was associated with a 0.28 unit increase in the index of reproduction among young unmarried women, after controlling for other demographic and socioeconomic factors. This indicates that FLE had an important influence upon the acquisition of knowledge/information, shaping of attitudes, and practices related to reproductive behaviour among young women.

Data also confirmed that access to FLE had a statistically significant and positive association with the index of contraception after controlling for other demographic and socioeconomic factors. For instance, a unit increase in FLE was associated with a 1.70 unit increase in the index of contraception confirming the substantial effect of FLE towards improving awareness, knowledge and benefits of different family planning methods among young unmarried women. Data also indicated statistically significant and positive association between access to FLE and index of HIV/AIDS, after controlling for other demographic and socioeconomic factors. For instance, a unit increase in FLE was associated with a 0.13 unit increase in the index of HIV/AIDS. This reinforces the fact that access to FLE helps to improve correct knowledge and understanding about HIV/AIDS among young women in India. However, the effect of FLE on the index of RTI/STI was statistically significant and negative, after controlling for other demographic and socioeconomic factors. For instance, a unit increase in FLE was associated with a 0.02 unit decrease in the index of RTI/STI among young women in India.

## 4. Discussion and conclusion

Using nationally representative household survey data, this study demonstrates the association between access to family life education and the range of experiences/awareness/attitudes related to sexual and reproductive health matters among young unmarried women aged 15–24 years in India. The study highlights the prevailing ignorance and inappropriate information/skills regarding an array of SRH matters among young unmarried women aged 15–24 years, who have never received any formal family life education. In addition, this research highlights stark demographic and socioeconomic differentials in access to FLE and awareness on a range of SRH matters among young unmarried women in India.

We report three key policy relevant outcomes that emerge from the analyses. First, the overall awareness and knowledge about range of sexual and reproductive health issues among

**Table 9. Multiple linear regression predicting sexual and reproductive health characteristics among unmarried young women (15–24y) in India, DLHS-3, 2007–2008.**

| Estimated regression coefficients parameter | Index of Menstruation | Index of Reproduction | Index of Contraception | Index of RTI/STI | Index of HIV/AIDS |
|---|---|---|---|---|---|
| | β-coefficient (95%CI) | β-coefficient (95%CI) | β-coefficient (95%CI) | β-coefficient (95% CI) | β-coefficient (95% CI) |
| **Intercept** | **1.12 (1.11,1.14)** | **1.41 (1.37,1.43)** | **1.70 (1.69,1.72)** | **-0.02 (-0.05, -0.00)** | **0.13 (0.09,0.16)** |
| **Received family life education** | | | | | |
| No | Ref. | Ref. | Ref. | Ref. | Ref. |
| Yes | **0.06 (0.05,0.07)** | **0.28 (0.26,0.29)** | **0.29 (0.28,0.30)** | **0.44 (0.42,0.45)** | **0.61 (0.59,0.62)** |
| **Age group (in completed years)** | | | | | |
| 15-19y | Ref. | Ref. | Ref. | Ref. | Ref. |
| 20-24y | **0.04 (0.03,0.04)** | **0.20 (0.18,0.21)** | **0.10 (0.09,0.11)** | **0.17 (0.16,0.18)** | **0.22 (0.21,0.24)** |
| **Residence** | | | | | |
| Rural | Ref. | Ref. | Ref. | Ref. | Ref. |
| Urban | **-0.01 (-0.02, -0.01)** | **-0.04 (-0.05, -0.02)** | **-0.01 (-0.02, -0.01)** | **0.04 (0.02,0.06)** | **0.09 (0.07,0.11)** |
| **Years of schooling** | | | | | |
| No education | Ref. | Ref. | Ref. | Ref. | Ref. |
| 1–5 years | **0.02 (0.01,0.03)** | **0.20 (0.18,0.23)** | **0.10 (0.09,0.12)** | **0.03 (0.01,0.06)** | **0.35 (0.32,0.39)** |
| 6–9 years | **0.06 (0.04,0.08)** | **0.43 (0.41,0.45)** | **0.21 (0.20,0.22)** | **0.16 (0.14,0.18)** | **1.10 (1.07,1.13)** |
| 10 years or above | **0.03 (0.01,0.05)** | **0.62 (0.59,0.64)** | **0.29 (0.28,0.31)** | **0.50 (0.48,0.53)** | **1.86 (1.83,1.90)** |
| **Employment status** | | | | | |
| Not working | Ref. | Ref. | Ref. | Ref. | Ref. |
| Agriculture | **-0.03 (-0.04,-0.02)** | **-0.03 (-0.09,-0.06)** | 0.00 (-0.01,0.01) | **0.02 (0.00,0.03)** | **-0.17 (-0.19,-0.15)** |
| Manual | **0.03 (0.02,0.04)** | **0.02 (0.01,0.05)** | **0.02 (0.01,0.04)** | **0.07 (0.05,0.09)** | -0.02 (-0.04,0.01) |
| Non-manual | **0.05 (0.04, 0.06)** | **0.11 (0.08,0.14)** | **0.09 (0.08,0.11)** | **0.22 (0.19,0.25)** | **0.10 (0.06,0.13)** |
| **Religion** | | | | | |
| Hindu | Ref. | Ref. | Ref. | Ref. | Ref. |
| Muslim | **0.05 (0.03,0.06)** | **-0.07 (-0.09, -0.06)** | **-0.04 (-0.05, -0.03)** | **-0.05 (-0.07,-0.04)** | **-0.27 (-0.29,-0.25)** |
| Christian | **0.04 (0.02,0.05)** | **0.16 (0.13,0.18)** | **-0.03 (-0.04, -0.01)** | **0.29 (0.26,0.31)** | **0.71 (0.68,0.74)** |
| Other religions | **0.09 (0.08,0.10)** | **0.18 (0.16,0.20)** | **0.02 (0.01,0.03)** | **0.20 (0.18,0.25)** | **0.49 (0.46,0.52)** |
| **Caste group** | | | | | |
| Scheduled Caste | Ref. | Ref. | Ref. | Ref. | Ref. |
| Scheduled Tribe | **0.01 (-0.00, 0.02)** | -0.01 (-0.02,0.01) | **0.02 (0.01,0.03)** | -0.01 (-0.03,0.00) | **0.09 (0.06,0.12)** |
| Other Backward Class | **-0.02 (-0.02,-0.01)** | -0.01 (-0.03,0.00) | 0.00 (-0.01,0.01) | **0.01 (0.00,0.03)** | **-0.04 (-0.06, -0.02)** |
| Other Caste | 0.00 (-0.00,0.01) | -0.01 (-0.02,0.01) | 0.00 (-0.01,0.01) | **0.01 (-0.01,0.02)** | 0.00 (-0.02,0.02) |
| **Wealth index** | | | | | |
| Lowest | Ref. | Ref. | Ref. | Ref. | Ref. |
| Second | **0.02 (0.01,0.03)** | **0.02 (0.01,0.05)** | **0.04 (0.03,0.05)** | **0.03 (0.01,0.05)** | **0.19 (0.17,0.22)** |
| Middle | **0.04 (0.03,0.05)** | **0.05 (0.03,0.07)** | **0.05 (0.04,0.06)** | **0.07 (0.05,0.09)** | **0.45 (0.42,0.48)** |
| Fourth | **0.08 (0.07,0.09)** | **0.11 (0.08,0.13)** | **0.04 (0.03,0.06)** | **0.16 (0.13,0.18)** | **0.71 (0.68,0.74)** |
| Highest | **0.10 (0.08,0.11)** | **0.16 (0.13,0.18)** | **0.05 (0.04,0.07)** | **0.27 (0.24,0.29)** | **0.88 (0.85,0.91)** |
| **State** | **0.00 (-0.00, 0.00)** | -0.01 (-0.01,0.01) | **-0.00 (-0.00, -0.00)** | **-0.00 (-0.00,0.00)** | 0.00 (0.00,0.00) |
| Adjusted R-square | 0.04 | 0.09 | 0.10 | 0.12 | 0.34 |

Bold indicates statistically significant at P<0.05.

young unmarried women in India was limited. Second, there were substantial demographic, socioeconomic and contextual differentials in awareness/attitudes regarding SRH matters among young unmarried women in India. Third, results demonstrate positive association

between training of family life education and attitudes regarding sexual and reproductive health behaviours among young unmarried women in India. Young women who had received any FLE were relatively better informed about issues of reproduction, contraception, transmission of RTI/STI and HIV/AIDS in comparison to their counterparts. Results from the multiple regression analyses confirmed that access to FLE was positively associated with the index of menstruation suggesting that better menstrual hygiene was practised by young unmarried women who received FLE in comparison to their counterparts [37–39]. Young unmarried women who received FLE were observed to have relatively higher index of reproduction, index of contraception and index of HIV/AIDS, confirming that provision of family life skills and learning could help the SRH matters of young unmarried women [40]. Interestingly, access to FLE were negatively associated with index of RTI/STI, suggesting that training related to family life education could be useful to reduce the likelihood of RTI/STI among young women.

These salient findings of the present study assume significance given the secular decline in the age at puberty among youth and the rising age at marriage across countries including India [41–43].These trends indicate the widening gap between the biological and social pubertal age (the age at which young people become mentally, educationally and legally prepared to function as adults in globalised modern world) of the adolescents/youth. These growing disparities between biological and social pubertal age leads to numerous public health concerns related to the lives of millions of adolescents and youth across the globe, particularly in developing countries including India. Puberty is generally associated with increased interest/fascination regarding sexual activities, a rise in curiosity and aggressive behaviour that stimulates risk-taking tendencies, often to conform with existing peer norms among adolescents and youth who constantly struggle to improve their social status in public life [44,45].These unfamiliar circumstances in both the biological and social spheres of life among adolescents and youth often push them into health damaging ill-informed risky behaviours including unprotected sexual intercourse with casual/multi-partners, exposure to RTIs/STIs, HIV/AIDS, unintended pregnancies, clandestine abortions, substance abuse, non-consensual sex, juvenile violence, bullying and suicidal attempts [7,41,46,47].

These results provide necessary evidence to tailor policies/programmes that appropriately train, guide and equip the adolescents/young people in India to develop essential skills about sexual and reproductive health matters in order to experience a smooth transition to adulthood [40]. Such initiatives may help to mentally/educationally/socially develop young people and also restrict the widening gap between biological and social pubertal age. Recognising these points, the Government of India tried to incorporate the family life/sex education program in the school curriculum during 2007, through the stewardship of the National AIDS Control Organization (NACO) and the Ministry of Human Resource Development (MHRD). However, this initiative received criticism from different strata of civil society/political leadership on the ground that it could malign/corrupt young, tender minds, promote promiscuity and denigrate the rich Indian culture etc., and was therefore withdrawn from various states across India [35]. Hence the programme of introducing sex education in the school curriculum was stopped before it had even started.

However, the ICPD+25 and other mission documents committed by the Government of India proclaim the right of adolescents/youth to be properly informed/trained about issues of SRH and the universal provision of SRH services to them. Given that Indian society is in transition, any discussion on sexual matters is considered taboo and sexual activity prior to marriage is stigmatised. Therefore, key stakeholders and leaders need to be taken on board during the programme design and implementation of FLE in Indian schools. Political will on the part of both the Union and state governments in India needs to strongly support the

implementation of the FLE programme by collaborating with programme planners, school principals, teachers, parents and others stakeholders to device appropriate and culturally conducive policies and programmes. This may also enable contemporary India to successfully reap the potential of their remarkable 'demographic *dividend*', which otherwise might become a '*missed demographic opportunity*'.

## Author Contributions

**Conceptualization:** Niharika Tripathi.

**Data curation:** Niharika Tripathi.

**Formal analysis:** Niharika Tripathi.

**Investigation:** Niharika Tripathi.

**Methodology:** Niharika Tripathi.

**Software:** Niharika Tripathi.

**Visualization:** Niharika Tripathi.

**Writing – original draft:** Niharika Tripathi.

**Writing – review & editing:** Niharika Tripathi.

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
