## [Decision Letter · Decision Letter 0]

26 Aug 2020

PONE-D-20-22730

Does Family Life Education Influence Attitudes Towards Sexual and Reproductive Health Matters among Unmarried Young Women in India?

PLOS ONE

Dear Dr. Niharika Tripathi,

Thank you for submitting your manuscript to PLOS ONE. After careful consideration, we feel that it has merit but does not fully meet PLOS ONE’s publication criteria as it currently stands. Therefore, we invite you to submit a revised version of the manuscript that addresses the points raised during the review process.

Kindly pay attention to the comments of the reviewers when reviewing the manuscript especially those that are related to methodological issues. 

Please submit your revised manuscript by 24th October 2020. If you will need more time than this to complete your revisions, please reply to this message or contact the journal office at plosone@plos.org. Please include the following items when submitting your revised manuscript:

We look forward to receiving your revised manuscript.

Kind regards,

Eugene Kofuor Maafo Darteh, Ph.D.

Academic Editor

PLOS ONE

'The funders had no role in study design, data collection and analysis, decision to publish, or preparation of the manuscript.'

4. Your ethics statement must appear in the Methods section of your manuscript. If your ethics statement is written in any section besides the Methods, please move it to the Methods section and delete it from any other section. Please also ensure that your ethics statement is included in your manuscript, as the ethics section of your online submission will not be published alongside your manuscript.

Reviewers' comments:

Reviewer's Responses to Questions

**Comments to the Author**

1. Is the manuscript technically sound, and do the data support the conclusions?

Reviewer #1: Partly

Reviewer #2: Partly

2. Has the statistical analysis been performed appropriately and rigorously? 

Reviewer #1: No

Reviewer #2: Yes

3. Have the authors made all data underlying the findings in their manuscript fully available?

Reviewer #1: Yes

Reviewer #2: Yes

4. Is the manuscript presented in an intelligible fashion and written in standard English?

Reviewer #1: Yes

Reviewer #2: No

5. Review Comments to the Author

Reviewer #1: This paper uses nationally representative data to assess exposure to family life education, and the association between exposure to family life education and knowledge about reproductive health issues (e.g., contraception, STIs, and HIV).

I have a few overall concerns about the analysis:

1. Knowledge of reproductive health outcomes was measured with several items. Why were the items treated as separate outcomes, instead of added for singles scores (i.e., one score on knowledge/awareness of contraception; one score on knowledge/awareness of HIV; one score on knowledge/awareness of STIs)? There is likely correlation among items in each set of questions.

2. It is unclear why family life education would be associated with menstrual problems (it is, but a justification for thinking that such education is related to such problems needs to be provided).

3. It is unclear if weights were used for the analysis. If not, why not?

Other comments

1. The abstract mentions two data sets, but the results are based on only one data set.

2. The introduction/background section can be streamlined, so as to focus on the importance of family life education to knowledge and reproductive health behaviors. I understand that improved reproductive health outcomes for young women may lead to other development outcomes, but that is beyond the scope of the analysis/paper.

3. The reproductive health knowledge variables need to be described in the text (it wasn't until I read the tables that I understood the analysis was looking at knowledge/awareness and not reproductive behaviors).

4. It is not clear, in table 1, whether any of the variables are significantly associated with exposure to family life education.

Reviewer #2: Thank you for the invitation to review this insightful paper. Though its of interest, there are essential issues that need to be addressed for it to be publishable.

General comments:

Authors should format the paper to the journal’s requirements. For instance, “Context” as used in the abstract section should be changed to “Introduction”. A thorough proofreading is required to rectify the typo and grammatical issues. There are several occasions where “adolescent” is used instead of “adolescents”, first sentence of third paragraph of introduction should bear “depict” not “depicts”, “unpleasantstate” “These includes” and several others.

Abstract

1. Under Context: revise to “…adolescents and young people…” not “…adolescent and young people…”

2. The author should remove “etc”

Introduction

3. Second paragraph: The author wrote “At many international negotiable platforms, including Programme of Action of the International Conference on Population and Development (ICPD, 1994), the World Plan of Action for Youth of the United Nations General Assembly (1995) and ICPD+15 2009, the global agenda was streamlined for empowering the situation of adolescents and youth [13–15]”. The conferences aimed at empowering the youth but not “empowering the situation of the youth” as indicated. This should be rectified.

4. This sentence should be referenced: “Majority of these youth reside in the poorest and least developed countries of Asia, Africa and Latin America which together constitute over 85 percent of world’s youth population.”

5. The author has done well by acknowledging some initiatives/policies in the country. Is there any data/evidence on how these have helped improved the attitude towards sexual and reproductive health?

6. The author wrote “India is one of the youngest nations…” what is the basis for this or is the author implying that it has a high proportion of young women/men globally?

7. The background needs to be summarised, it is too bulky. The historical accounts can be truncated and rather project the Indian context.

Methods

8. One of the outcome variables is “menstruation related problems”. Most of the problems are biological and conceptually does not make sense to investigate with Family Life Education.

9. How was “family life education” as an outcome variable measured and what does it constitutes? Explanation is required here.

10. Change “Methods” to “data analysis”

Results

11. The author can present about three regression results per table to make it more presentable.

Discussion and conclusion

The author spends more time describing other studies in some instances instead of situating the present study within the existing literature. For instance, this occupies a full paragraph but not informative enough for a discussion “Another household level representative survey entitled ‘Youth in India: Situation and Needs’ was conducted in 2006-07 across six Indian states [26]. The main objective of this survey was to gather evidence on key life transitions experienced by youth as well as their awareness, attitudes and life choices. The study was conducted in the following selected Indian states: Andhra Pradesh, Bihar, Jharkhand, Maharashtra, Rajasthan and Tamil Nadu. Overall, about 50,773married and unmarried young women and men were successfully interviewed, from 1,74,037 sample households. Unmarried men and women as well as married women (15-24 years) were interviewed, whereas the age group for married men was extended to 15-29 years, in the first ever 22 landmark survey on youth in India. The results from this survey lend support to the findings of our study”

6. PLOS authors have the option to publish the peer review history of their article (what does this mean?). If published, this will include your full peer review and any attached files.

Reviewer #1: No

Reviewer #2: No

---

## [Author Response · Author response to Decision Letter 0]

14 Oct 2020

We thank the editor of PLOS ONE and two reviewers for their valuable comments and suggestion that really helped us to improve the manuscript. Detailed point by point reply to the reviewer's comments has been attached with the manuscript.

---

## [Decision Letter · Decision Letter 1]

27 Nov 2020

PONE-D-20-22730R1

Does Family Life Education Influence Attitudes Towards Sexual and Reproductive Health Matters among Unmarried Young Women in India?

PLOS ONE

Dear Dr.Niharika Tripathi,

Thank you for submitting your manuscript to PLOS ONE. After careful consideration, we feel that it has merit but does not fully meet PLOS ONE’s publication criteria as it currently stands. Therefore, we invite you to submit a revised version of the manuscript that addresses the points raised during the review process.

I would not treat the measures of knowledge as individual items in bi-variate analysis.  It isn't clear that the level of detail is warranted.  Also, kindly address the comments of the review below. The paper needs copy editing.

We look forward to receiving your revised manuscript.

Kind regards,

Eugene Kofuor Maafo Darteh, Ph.D.

Academic Editor

PLOS ONE

Reviewers' comments:

Reviewer's Responses to Questions

**Comments to the Author**

1. If the authors have adequately addressed your comments raised in a previous round of review and you feel that this manuscript is now acceptable for publication, you may indicate that here to bypass the “Comments to the Author” section, enter your conflict of interest statement in the “Confidential to Editor” section, and submit your "Accept" recommendation.

Reviewer #1: (No Response)

Reviewer #2: All comments have been addressed

2. Is the manuscript technically sound, and do the data support the conclusions?

Reviewer #1: Partly

Reviewer #2: Yes

3. Has the statistical analysis been performed appropriately and rigorously? 

Reviewer #1: Yes

Reviewer #2: Yes

4. Have the authors made all data underlying the findings in their manuscript fully available?

Reviewer #1: Yes

Reviewer #2: No

5. Is the manuscript presented in an intelligible fashion and written in standard English?

Reviewer #1: Yes

Reviewer #2: Yes

6. Review Comments to the Author

Reviewer #1: I appreciate the revisions to the manuscript. I have a few remaining comments.

1. In the methods section, you indicate that you created scales to measure knowledge and awareness of RH matters. In fact, you treat the variables as individual items in the bi-variate analysis and as scales in the multi-variate analysis. I would treat them as scales in both the bi-variate and multi-variate analysis. If not, you need to clarify that you treat the variables in two different ways.

Reviewer #2: Thanks to the authros for addressing all my initial comments. I have no further comment on this manuscript.

7. PLOS authors have the option to publish the peer review history of their article (what does this mean?). If published, this will include your full peer review and any attached files.

Reviewer #1: No

Reviewer #2: No

---

## [Author Response · Author response to Decision Letter 1]

3 Dec 2020

We thank the editor and reviewer for their valuable comments. We have addressed them in the text of the manuscript and indicated in the rebuttal letter to the reviewer.

---

## [Decision Letter · Decision Letter 2]

11 Jan 2021

Does Family Life Education Influence Attitudes Towards Sexual and Reproductive Health Matters among Unmarried Young Women in India?

PONE-D-20-22730R2

Dear Dr. Niharika Tripathi,

We’re pleased to inform you that your manuscript has been judged scientifically suitable for publication and will be formally accepted for publication once it meets all outstanding technical requirements.

Kind regards,

Eugene Kofuor Maafo Darteh, Ph.D.

Academic Editor

PLOS ONE

Additional Editor Comments (optional):

Reviewers' comments:

Reviewer's Responses to Questions

**Comments to the Author**

1. If the authors have adequately addressed your comments raised in a previous round of review and you feel that this manuscript is now acceptable for publication, you may indicate that here to bypass the “Comments to the Author” section, enter your conflict of interest statement in the “Confidential to Editor” section, and submit your "Accept" recommendation.

Reviewer #1: All comments have been addressed

Reviewer #2: All comments have been addressed

2. Is the manuscript technically sound, and do the data support the conclusions?

Reviewer #1: Partly

Reviewer #2: Yes

3. Has the statistical analysis been performed appropriately and rigorously? 

Reviewer #1: Yes

Reviewer #2: Yes

4. Have the authors made all data underlying the findings in their manuscript fully available?

Reviewer #1: Yes

Reviewer #2: No

5. Is the manuscript presented in an intelligible fashion and written in standard English?

Reviewer #1: Yes

Reviewer #2: Yes

6. Review Comments to the Author

Reviewer #1: I have no further comments for the author at this point, and have made my recommendation in other parts of this form.

Reviewer #2: I have no additional comment for the authors. Thanks to the authors for attending to the comments I raised.

7. PLOS authors have the option to publish the peer review history of their article (what does this mean?). If published, this will include your full peer review and any attached files.

Reviewer #1: No

Reviewer #2: No

---

## [Editor Report · Acceptance letter]

15 Jan 2021

PONE-D-20-22730R2 

Does Family Life Education Influence Attitudes Towards Sexual and Reproductive Health Matters among Unmarried Young Women in India? 

Dear Dr. Tripathi:

I'm pleased to inform you that your manuscript has been deemed suitable for publication in PLOS ONE. Congratulations! Your manuscript is now with our production department. 

Kind regards, 

on behalf of

Dr. Eugene Kofuor Maafo Darteh 

Academic Editor

PLOS ONE